# Polymer Retention Determination in Porous Media for Polymer Flooding in Unconsolidated Reservoir

**DOI:** 10.3390/polym13162737

**Published:** 2021-08-16

**Authors:** Ilnur Ilyasov, Igor Koltsov, Pavel Golub, Nikolay Tretyakov, Andrei Cheban, Antoine Thomas

**Affiliations:** 1JSC “Messoyakhaneftegaz”, Holodilnaya Street, 77, 625026 Tyumen, Russia; 2JSC “Gazpromneft—TP”, 50 Years of October Street, 14, 625048 Tyumen, Russia; koltsovi@mail.ru; 3Coretest Service, Co., Ltd., Lenina Street, 2a, 625003 Tyumen, Russia; golub_pp@coretest.ru (P.G.); andre-ch2008@yandex.ru (A.C.); 4Research Resource Center “Natural Resource Management and Physical-Chemical Research”, Institut of Chemistry, University of Tyumen, Volodarskogo, 6, 625003 Tyumen, Russia; n.y.tretyakov@utmn.ru; 5ICUBE Laboratory, UMR 7357 CNRS, University of Strasbourg, 67000 Strasbourg, France; 6Institute of Physics and Technology, University of Tyumen, 625003 Tyumen, Russia; 7141 Avenue Frères Lumière, 69008 Lyon, France; antoinethom@gmail.com

**Keywords:** polymer dynamic retention, adsorption, unconsolidated core, polymer flooding, non-absorbable tracers, viscous oil

## Abstract

Polymer flooding is a well-established technique aimed at improved recovery factors from oilfields. Among the important parameters affecting the feasibility of a large deployment, polymer retention is one of the most critical since it directly impacts the oil bank delay and therefore the final economics of the project. This paper describes the work performed for the East-Messoyakhskoe oilfield located in Northern Siberia (Russia). A literature review was first performed to select the most appropriate methodology to assess polymer retention in unconsolidated cores at residual oil saturation. 4 polyacrylamide polymers were selected with molecular weights between 7 and 18 M Da and sulfonated monomer (ATBS) content between 0 and 5% molar. An improved 2-fronts dynamic retention method along with total organic carbon—total nitrogen analyzers were used for concentration measurement. Retention values vary between 93 and 444 The sentence could be rephrased μg/g, with the lowest given by the polymers containing ATBS, corroborating other publications on the topic. This paper also summarizes the main learnings gathered during the adaptation of laboratory procedures and paves the way for a faster and more efficient retention estimation for unconsolidated reservoirs.

## 1. Introduction

Among the various existing enhanced oil recovery (EOR) techniques implemented in the world and described in the international literature, polymer flooding is probably one of the most established. It basically consists in increasing the viscosity of the injected water to improve and accelerate oil displacement in reservoirs where heterogeneity is present and/or the mobility contrast between the oil and the injected water is significant [1]. Despite the relative simplicity of the concept, several factors condition the success of any implementation. One of them is injectivity which, described in broad terms, defines how much polymer can be injected, at which viscosity and rate, to recover oil in a timely and economic fashion. A second one is polymer retention describing how much polymer is lost in the reservoir during the injection, parameter which also directly impacts injectivity and therefore the success or failure of the project. The focus of this paper will be on the second point i.e., retention and the benchmarking of several polymers for a deployment in the East-Messoyakhskoe oil field in Siberia, Russia. More details regarding the oil field, the reservoir characteristics and the pilots’ results can be found elsewhere [2,3].

The retention study described herein was part of a comprehensive project designed to assess the feasibility of this technique to improve the oil recovery factor in the aforementioned oilfield. At first, it seemed to be a simple task but quickly several challenges arose including the use of unconsolidated rocks, their restoration and measurement of all parameters related to polymer in the presence of oil: this is described in the present paper. To start with, the choice of a suitable polymer was conducted by comparing the performance of several chemicals in regard to the following criteria:Viscosifying power: the polymer should provide the required in-situ viscosity (more precisely resistance factor) at a reasonable concentration at reservoir conditions;Retention: polymer loss in the reservoir should be as low as possible to minimize propagation issues and oil bank delay. This is the focus of this study;Injectivity: the polymer solution should propagate easily through the reservoir rock. This factor is directly impacted by viscosity and retention as well as the chemical composition of the polymer;Long-term stability: the polymer should be stable and provide enough viscosity throughout its transit in the reservoir;Cost.

In that process, the evaluation of retention took a critical place to rank the candidates since the reservoir conditions are quite favorable for the use of conventional polyacrylamide chemistries. Indeed, the reservoir temperature (16 °C) and injection water salinity (16 g/L) are low which favors a good polymer stability and the obtention of the target viscosity at reasonable polymer concentration (0.01 Pa·s) @ 900 mg/kg commercial concentration). Retention, on this other side, also depends on the mineralogy and other reservoir characteristics such as permeability and porosity. It is really a critical parameter since large polymer losses through retention can dramatically jeopardize the process and delay the oil bank [4]. Sensitivity analyses can be carried out through simulation using the values obtained during the laboratory experiments to draw a business case and de-risk the feasibility of injection in the field.

Despite large and numerous projects overseas, polymer flooding remains a relatively new tool in the shed of Russian oil companies with very few published cases [5,6] with a scale comparable to what has been implemented in Canada [7], Argentina [8,9,10] or India [11,12,13], to name but a few. Moreover, for East-Messoyakhskoe, the fact that the field is located in subarctic conditions makes the development of EOR techniques even more challenging. The recent and very rapidly growing renewed interest for chemical enhanced oil recovery techniques in Russia also poses an exciting challenge: the skills and equipment needed to carry out comprehensive laboratory tests are currently being gathered and, therefore, this paper also intends at summarizing the best practices and skillsets, especially for the Russian operators who are willing to study the feasibility of injecting polymer to improve the recovery factor for their oilfields.

In the next paragraph, we will give a brief overview of procedures related to retention studies and the challenges faced in this case.

## 2. Determination of Retention: Review of Procedures and Challenges

Polymers used in EOR are macromolecules which interact with the rock during their transit through the oil-bearing formation and, as such, a certain percentage of these molecules can be lost by retention. The term “retention” encompasses 3 mechanisms [14]:Adsorption: the polymer molecules can “stick” to the rock via Van der Waals, ionic or hydrogen bonds. Adsorption is usually considered irreversible since the macromolecule can attach to the rock via many points;Mechanical entrapment: if the hydrodynamic volume of the molecule in solution is larger than the pore throat it encounters, then the polymer molecule can become physically trapped at the entrance of the pore;Hydrodynamic retention: in some cases, the molecules can be (temporarily) trapped in eddies or regions where the flow is stagnant.

Retention values are expressed in micrograms of polymer lost per gram of rock. In the literature, values ranging from 9 to 700 μg/g (micro grams of polymer per gram of rock) are reported for various polymers and rocks. Indeed, large losses by retention can cause significant delay in oil production. It is possible to calculate such delay for various injected concentrations and retention values using Formula (1):V_ret_ = [ρ_rock_(1 − φ)/φ] × [R_pret_/C_polymer_](1)

With V_ret_ the pore volume delay, ρ_rock_ the rock density, φ the porosity in %, R_pret_ the polymer retention in μg/g and C_polymer_ the polymer concentration in parts per million. Considering a density of 2.65 g/cm^3^ and a porosity of 25%, it is possible to draw a simple graph showing the pore volume delay for various polymer concentrations and retention values.

From Figure 1, for low polymer concentrations, any loss by retention becomes rapidly problematic and can render the project uneconomic. This is also a reason why choosing the polymer providing the highest resistance factor at the lowest concentration to save on operational expenditures can become a problem if the concentration is low and retention occurs. For example, at 500 mg/kg, a retention value of 30 μg/g leads to a delay of 50% pore volume which clearly compromises the success of the injection.

In the case at East-Messoyakhskoe, the target polymer viscosity is reached for a concentration of 900–1300 mg/kg commercial, depending on the polymers considered. Therefore, a sensitivity analysis must be carried out to calculate the possible delay and attempt to mitigate losses by retention.

Several families of procedures exist to quantify polymer retention/adsorption, the first one covering static methods [15,16] and the second one covering several dynamic methods [17,18]. The static method generally uses crushed rock samples mixed with a polymer solution of known concentration. After soaking, a centrifugal process isolates the solid from the liquid and the remaining polymer concentration in solution is determined by titration. The adsorption is then calculated by difference. This approach has several important limitations. Firstly, the area available for the polymer to get in contact with the rock is much higher than inside a consolidated or unconsolidated reservoir. Secondly, it is not possible to assess the impact of oil on retention.

The second approach, called dynamic, considers the injection of one or several polymer fronts with a tracer [17,18]. In the case of 2 fronts with tracer, the retention is determined by the area between both normalized polymer curves, at the 0.5 cut-off of the ratio (concentration effluents/initial concentration). The tracer is used to determine inaccessible pore volume (IAPV) (Figure 2). In this paper, two methods were compared: the two fronts described in Figure 2 and another consisting in considering only the first polymer and tracer fronts and measuring the respective concentrations during build-up and water flush (Figure 3).

At the very beginning of the project, the static method was used to compare polymers and obtain retention values. The results obtained appeared very high compared to other cases reported in the literature. As experience grew, it was decided to investigate other techniques to assess retention in a manner closer to what could be expected in the field. A literature review indicated that a dynamic method with double front injection along with tracers was a more accurate and widely used method.

Several authors have studied the different factors affecting the retention of polyacrylamides and possible mechanisms behind it, a short list of which has been summarized below. Among the observations that impacted how this study was conducted, we highlighted the following:Polymer concentration can have a significant impact on retention, especially when displacing viscous oils [18].Polymer retention values can be 2 to 10 times lower in the presence of oil than without oil [19].The presence of sulfonated monomers in the polymers (ATBS) can help decrease the retention of polymers [20].The mineralogy of the reservoir, and especially clay and iron content, greatly impacts polymer retention [21].Polymer characteristics such as chemical composition, concentration, and molecular weight impact retention [22].Retention values vary if the experiments are conducted in aerobic vs. anaerobic conditions [23].The packing and ageing methodology have a significant impact on the overall results [24].Retention can be flow rate dependent, especially in low permeability rock [16,25].The correlation between retention values and inaccessible pore volume in the core is still unclear [18].

Consequently, for the retention study for East-Messoyakhskoe oilfield, it was decided to proceed as follows:The dynamic retention method with 2 polymer fronts was used as a standard.The method was improved for unconsolidated rocks and technical abilities of the laboratory which conducted the tests.The retention values were obtained in cores at residual oil saturation, to mimic real field conditions.The retention studies compared polymers at iso-viscosity (0.01 Pa·s) to get as close as possible from the injection conditions.Effluent concentration of polymers was determined via total nitrogen concentration (TNb method).Polymers with different molecular weights and chemistries (with or without Acyrlamido Tertiary Butyl Sulfonate or ATBS) were compared to determine the retention trends.Core with different permeabilities were used to represent the heterogeneity in the field.

As for concentration measurements, several literature articles and standards were revised to find analytical methods which could be useful to determine polymer concentration in effluents from laboratory corefloods and field EOR operations. These methods are necessary to determine how much of these chemicals is retained inside porous media.

Currently, six methods are referenced in the literature to determine polymer concentrations when exploiting the polymer or surfactant-polymer flooding technique:Method 1: Rheological determination using an isotorque curve [26].Method 2: Improved starch-iodide method [27].Method 3: Flocculation test [28].Method 4: Colloid titration [29].Method 5: Improved bleach method [30].Method 6: Total Organic Carbon matter content determination TOC [31].

Although all analytical methods mentioned above can be used to determine polymer concentrations, they have different accuracy and sensitivity. The elemental analysis method based on the measurement of total organic carbon (TOC) or total nitrogen (TNb) has the advantage that it is accurate and easy to perform while enabling discriminating between carbon from the polymer and from the oil present in the rock. Instruments commercially available can determine both TOC and total nitrogen (TNb), automatic mode included. Such equipment comes handy and convenient to determine polymer in filtration test samples when dozens or hundreds of individual fluid samples need to be analyzed. This technique was used for this study.

## 3. Materials

### 3.1. Core

Experiments were carried out using core samples from the East Messoyakhskoe field, PK1-3 reservoir (name of formation), permeability in that reservoir varying widely, from 70 to 15,000 × 0.987 × 10^−12^ m^2^ (or millidarcy, md).

To reproduce the heterogeneous rock structure in the laboratory model, two groups of samples with different permeability classes were selected to represent field heterogeneity. The first group of samples, with a helium permeability of 300–700 md, simulated a zone of low to medium reservoir permeability (20% of facies). The second group of samples, with helium permeability of 1150–1850 md, simulated the medium-permeability zone of the formation (40% of the facies). However, while preparing the samples, it was found that core tests using the actual core of the weakly-consolidated reservoir of interest (named PK 1–3) were only possible for permeabilities not higher than 1.5 d: samples tend to fracture at higher permeabilities (40% of the facies). Individual stable samples of higher permeability may be found, but it is impossible to consolidate them into a column of homogeneous properties.

It was important to bring samples to their natural state. For that purpose, cylindrical core samples were produced, extracted, porosity and permeability properties were determined under atmospheric conditions. These samples were first saturated with synthetic formation water, and then with reservoir oil, bringing the rock at residual water saturation. A total of 6 composite models of 5 samples each were prepared, the conditions of experiments are described below.

### 3.2. Polymers and Solutions

The composition of the brine used is described in Table 1. The polymer solution was prepared using synthetic water with mineral composition as close to formation water as possible.

Four types of anionic polyacrylamide polymers of different molecular weights featuring different additives were used, the summary of experimental conditions is shown in Table 2. The molecular weight mentioned in the table corresponds to the mean considering a gaussian distribution for the polyacrylamide samples (in million Dalton, MDa or million g/mol).

Before testing, the polymer concentration was determined to reach a target viscosity of 0.01 Pa·s at 25 °C. Viscosity was determined using a Brookfield DV2T viscometer with UL adapter at shear rates of 7.3, 14.6 and 21.9 s^−1^. Usually, polymer retentions are compared at their equal concentrations since the latter has an impact on retention. In our study, we purposely chose to compare them at equal (target) viscosities, as this method is closer to the practical engineering goals at this stage of polymer comparison, especially after a target viscosity has been chosen.

## 4. Methods

### 4.1. Set up of Filtration Test

Polymer compositions were tested on the experimental Tech-ViP unit produced by TECH-INTENSIVE LLC. The hydraulic diagram of the unit is shown in Figure 4.

This unit reproduces thermobaric conditions close to those in the formation and is designed for a pore pressure of 25 MPa, a rock pressure of 60 MPa and a temperature of up to 110 °C, which fully meets the requirements of the experiment given in Table 3.

An isoviscous model was used to simulate reservoir oil. To prepare it, degassed oil was diluted with a solvent (petroleum ether) to match the oil viscosity at reservoir conditions. Before preparing the model, degassed oil was purified to filter out mechanical impurities, water, and solid phase.

Since the purpose of the study was to obtain data as useful as possible from the engineering point of view, retention was performed in cores at residual oil saturation. The presence of hydrocarbons in the core enables approximating the actual formation conditions during the filtration experiment but creates additional difficulties in calculations and in choosing the technique to measure the polymer content.

### 4.2. Determination of Polymer Retention

The experiment methodology corresponded to generally accepted approaches, where formation water was pumped through a composite core model of initial oil saturation (*K*_io_) and residual water saturation (*K*_rw_) to create residual oil saturation (*K*_ro_), thus simulating the 2nd stage of field development. Next, simulating the 3rd stage of field development, several pore volumes of polymer solution were sequentially pumped through that composite core model now having its residual oil saturation (*K*_ro_) (first front); then it was washed with simulated formation water and pumped with the polymer solution again (second front).

However, taking all the above into account, we concentrated on the following aspects of poorly cemented core to shape our methodology:Testing of polymer compositions should be performed on composite models of 5 cylindrical core samples having permeabilities of class 1 of clastic reservoirs (helium permeability of 1150–1850 md) and class 2–3 (helium permeability of 300–700 md) to simulate different sections of the reservoir. It was not possible to cover the whole permeability range, mentioned above, especially with high permeability more than 1.5 d.Fix weakly cemented specimens with PTFE film (polytetrafluoroethylene), without a brass mesh at the ends: using a mesh is not recommended as it can affect polymer retention at the ends of the specimens.Samples shall be cleaned in a Soxhlet apparatus with an alcohol and benzene mixture. The core shall be dried at a temperature not exceeding 70 °C to prevent decomposition of clay minerals.Samples shall be saturated with synthetic multicomponent water, its composition being as close to that of formation water as possible.Forming the residual water saturation (*K*_rw_) and initial oil saturation (*K*_io_), restoring wettability of samples and all subsequent preparations shall be performed in a filtration unit to avoid a mechanical impact and destruction of core samples.At all stages of testing, the linear fluid injection rate should not exceed 2 m/day to ensure reproducing the actual filtration rate and capillary impregnation in actual development of the formation under study.More exact determination of optimum polymer slug size, formation water model, and pumping rates. Slug sizes are selected based on the heterogeneity of the reservoir, oil to polymer solution viscosity ratio and the degree of polymer adsorption to pore space. The optimum pumping volumes and rates were experimentally determined during the first experiment, as shown in Table 4.

### 4.3. Detection of Polymer Concentration

The polymer concentration was determined by elemental analysis, estimating the total nitrogen concentration (TNb) on a Vario TOC Cube, Elementar. The elemental analyzer was graduated using polymer solutions with exactly known concentration, prepared according to the same procedure (Figure 5). Each graduation solution was analyzed three times, and the graduation dependence was plotted against the average response of the electrochemical detector of the analyzer (taking the blank analysis of model formation water into account). 400 µL (micro liters) of polymer or graduation solution is then injected into the analyzer reactor.

The liquid sample enters the combustion furnace where all carbon-containing components are oxidized by purified air in the presence of a catalyst to form carbon dioxide, which is detected by an infrared detector. All polymers used in the study contain nitrogen in their molecules. In this case, it is converted into nitrogen monoxide during oxidation in the furnace in the presence of a catalyst, the nitrogen monoxide being registered by an electrochemical detector. By connecting in series, the carbon dioxide and nitrogen monoxide detectors, TOC and TNb respectively, can be determined simultaneously with a single injection of a sample into the analyzer.

Since oil-saturated core was used during filtration tests, some oil components may dissolve in water during the sampling process. In that case, the result of measuring organic carbon will be overestimated. Therefore, polymer concentrations were calculated from total nitrogen in all tests.

Samples of polymer solution passed through the core column were centrifuged for 15 min at 3000 rpm (rotations per minute). A 1000 µL aliquot of the aqueous phase was transferred to a 25 mL volumetric flask and diluted with distilled water. The diluted sample was placed in a test tube for automatic introduction into the analyzer. Each sample was analyzed three times. Detector graduation enabled retrieving linear dependences to calculate polymer concentrations.

### 4.4. Determination of Tracer Concentration

Usually, researchers use inorganic salts as indicators. But salt addition leads to a change in the ionic strength of the solution, which can reduce the viscosity of the polymer solution. Since in our case the comparison was made under the conditions of equal viscosity of 0.01 Pa·s, the preferred method is the one in which the viscosity remains constant and does not depend on any additional factors. Therefore, low-molecular-weight alcohols were used as the tracer in this study. Partitioning coefficients in the heavy oil were found to be very low (below 1%). Alcohol concentration was determined by gas chromatography (using the Chromatec-Crystall 5000.2 instrument). The detector was a flame ionization unit. The detector was graduated using standard alcohol solutions in model formation water in the range of 50–500 ppm. An individual tracer with a concentration of 250 mg/kg was injected into the polymer solutions.

## 5. Results and Discussion

### 5.1. Coreflooding Results

During the experiments (the sequence is summarized in Table 4), pressure drop across the central sample and across the entire core column was recorded, plus samples between 0.1 to 0.5 pore volumes were taken at the outlet, depending on the stage of the experiment, to determine the concentration of the polymer in the effluents. The results of test No. 6 are shown in Figure 6.

On Figure 6, we can see both polymer fronts separated by 50 pore volumes of water to clean the core in between and remove non-adsorbed polymer. Two pressure curves are presented corresponding to the center of the core (orange/cross) and the full model (blue/dots). During the first injection, some oil has been produced, translating into “bumps” on both curves, impacting the pore volume value initially determined. Overall, the injectivity of polymers appears good since the pressure plateaus for both fronts.

Based on the results of these experiments, graphs were plotted similar in the format and in the meaning to Hall curves. Hall curves are commonly used in the field to estimate the processes that affect changes in the well or formation injectivity from the well pressure dynamics and the volume of injected fluid [32]. Such a graph displays the multiplication of cumulated pressure (*p*) with time increment (d*t*) (y-axis) as a function of cumulated injected volume (x-axis). In case of a laboratory filtration test, the accumulated volume of injected fluid is plotted along the ordinate axis, and the accumulated pressure is plotted along the abscissa axis. When operating with the accumulated pressure and injection volume, minor fluctuations level off and significant changes in fluid injection (flow) patterns become visible.

Figure 7 shows the adapted Hall plots for experiments No. 2 and 6. The diagnostic parameter is the deviation of the graph from linearity. In this case, the deviation can have both a positive effect on the injectivity increase (the slope of the graph decreases) and a negative effect (the deviation from linearity increases and is directed upwards). The graph shows that when injecting a more viscous polymer front, the accumulated pressure increases, i.e., the slope of the curve becomes steeper. The opposite phenomenon has also been observed, where the curve becomes less steep during the post-flooding periods. Note that inflection points of the graph agree well with the start and the end of the polymer pumping front.

Interestingly, the Hall plots from experiments No. 2 and 6 are markedly different. In the latter case, the curve is considerably flatter. It means that significantly less accumulated pressure is required to pump an equal volume of fluid into the formation. That is primarily due to a higher permeability of core sample and the use of a polymer of lower molecular weight. Recall that the volumetric viscosity of the two solutions was equal, but equal volumetric viscosities of polymer solutions do not guarantee equal effective viscosities in a porous medium. Such a comparison of the two filter tests and the respective figure reflects this rule well. Overall, using an analog approach to the Hall plot for cores helps understand the injection dynamics and verify the good injectivity of the polymer solution. This method can be used as an effective and quick screening tool to rank several polymers.

### 5.2. Polymer and Tracers Measurements

The equilibrium concentration of polymer in the samples was calculated using the corresponding linear equation (Figure 5). For all the polymers used, the linear regression coefficient stayed within 0.9817–0.9998. While determining the polymer concentration, graduation solutions were analyzed regularly. An example of an output curve with two polymer fronts and tracers is shown in Figure 8. The normalized tracer and polymer concentrations are plotted on the overall solution injection graph, where the injected volume of fluid is plotted along the abscissa axis in dimensionality of the pore volume of the sample, sometimes in ml. In our case, each polymer slug has its own tracer. A careful comparison is then made between polymer profile once steady state is reached and the corresponding tracer. It should be emphasized that the adsorption (retention) of polymer in the first front must reach saturation before pumping the second front to obtain accurate results.

Even a plain visual analysis of the presented curve entails several conclusions. The first black peak (black diamond curve) in the figure shows an increase and decrease in concentration of non-sorbable tracer No. 1 in aqueous medium. A similar non-sorbable tracer was used with the first polymer front (dark blue line with squares). One may see that the rate of reaching the maximum concentration of 100% (the normalized value in the *C*_eff_/*C*_i_ = 1 plot) does not depend on whether the tracer propagates in water or with the polymer front. The slopes of the gray and dark blue lines remain constant. Meanwhile, the advance of the polymer front (orange line, triangles) significantly lags behind the non-sorbable tracer due to retention of the polymer.

The fluids under study, including water and the indicator (tracer), can pass through the entire pore volume of the porous medium. In addition, there is often a portion of the pore volume into which polymer molecules cannot penetrate. This volume is referred to in the literature as the inaccessible pore volume (IAPV) [18,24]. Inaccessibility of a part of pores for the polymer is usually associated with the large size of macromolecules and their elastic deformation.

Estimating the inaccessible pore volume is necessary both to clarify the value of polymer retention in the formation and to build hydrodynamic models of polymer flooding. When interpreting the obtained tracer and polymer production curves, the IAPV was calculated, the average value of which was 2–3%. The low IAPV value is caused by high porosity and permeability of weakly consolidated core of PK1-3 formation, which is a favorable factor for implementing polymer flooding.

### 5.3. Polymer Retention Values

The results obtained during the studies were consolidated in a summary Table covering all the experiments. Then they were analyzed, and retention values were calculated using the dynamic method with increased injection and two polymer fronts (Method 1) and the concentration profile method (Method 2) described above. Summary results are shown in Table 5.

The following conclusions can be drawn from these results, by comparing what can be or by drawing trends:The retention values are similar to other cases described in the literature for unconsolidated heavy oil formations.Retention decreases with the increase in permeability and the decrease in molecular weight.However, it was not possible to measure retention for permeabilities more than 1.2 d, which represents the majority of permeability range. So, the retention values in the reservoir should be much lower.The addition of ATBS in the polymer macromolecule reduces polymer retention in the studied cores. For instance, a 12 MDa (million g/mol) polymer with ATBS displays similar retention values than a 7 MDa conventional polyacrylamide in rocks with comparable permeabilities.Discrepancies exist between both measurements’ methods with reasons still unclear at this stage. Investigations are on-going to get a better understanding of these results.

More polymers should be tested in a variety of other reservoir conditions but the number of available samples, time, and money force the adaptation of test procedures to draw usable trends. Clearly, in this case, to maximize the chance of success and increase the sweep efficiency in the reservoir taken as a whole, a polymer with a medium molecular weight containing ATBS should be considered. Economic calculations will be done to determine the optimal polymer choice considering retention, concentration needed to reach the target viscosity and the extra cost of adding a monomer such as ATBS.

### 5.4. Discussion

Even though there are many publications on techniques used to estimate polymer retention in a formation, the practical application of these techniques for engineering purposes has its limitations. These limitations are often due to lack of time for experiments and limited core material. As a rule, there is no possibility to conduct time-consuming preliminary tests to adapt the technique to specific conditions. In addition, unconsolidated cores in combination with viscous formation oil have their specific features and require a special approach and adaptation of methods.

In the current experiment, a significant pressure increase on the composite core column as well as dissimilarities in pressure differences at the model ends and at its center were noted. After injection of 3 pore volumes of polymer, the initial polymer concentration was not reached at the column outlet; in other words, the polymer was adsorbed somewhere inside the core column. Each of these observations taken separately could be interpreted in various ways but taken in combination they indicate that polymer adsorption takes place predominantly at the inlet end of the core column. To avoid that effect, we decided to change the injection rate and volume. This highlights the need for internal pressure taps to properly assess the polymer propagation in the core.

Another important point was the use of a different interim tracer which eliminates possible determination errors associated with long-time tracer going out during displacement and superimposition on the tracer from the second polymer front. Moreover, confidence that the laboratory will easily distinguish two tracers allowed additional reduction in the volume of water washing between two fronts, which decreases the total duration of the experiment (but still flushing more than 50 pore volumes with water).

We also confirmed that the influence of mineralogical composition and claystone consistency affects the results too. On the one hand, this confirms the methodology to be functionally adequate. On the other hand, when planning future experiments, it will be necessary to select samples of similar mineralogical composition and to perform XRD analysis.

Finally, despite the need for further investigations about the discrepancies observed for both methods, this study tends to confirm the applicability of the developed methodology to determine polymer retention in laboratory studies of unconsolidated viscous oil reservoirs.

## 6. Conclusions

A comprehensive methodology for determining polymer retention was developed for unconsolidated reservoirs with viscous oils. The technique was adapted to the technical capabilities of the laboratory and tested using an actual core from PK 1–3 formation of the East-Messoyakhskoe field. After adapting the existing methods to the specificities of this case and the available equipment, it was decided to characterize polymer retention in reservoir cores at residual oil saturation.

Four polymers with different characteristics were tested. All were polyacrylamide polymers with molecular weights ranging from 7 to 18 MDa with, for 2 of them, the presence of a sulfonated monomer known to decrease polymer retention (ATBS).

Retention values were determined using dynamic methods consisting in the injection of polymer with tracers. The values obtained range from 93 µg/g to 444 µg/g on average, with the lowest value obtained for a polymer containing ATBS and/or high permeability rocks. These values are somewhat like other published cases for unconsolidated heavy oil reservoirs. 

However, it was not possible to measure retention for permeability more than 1.2 d, which represents the majority of permeability range. So, the retention values in the reservoir should be much lower.

They will be used for simulation purposes to better define a business case and the feasibility of polymer full field injection in this oilfield.

During this work, a cross-functional team was formed, great experience was gained which resulted in new complex procedures to determine polymer retention with direct application to the unconsolidated heavy oil PK 1–3 reservoir of East-Messoyakhskoe field.

## Figures and Tables

**Figure 1 polymers-13-02737-f001:**
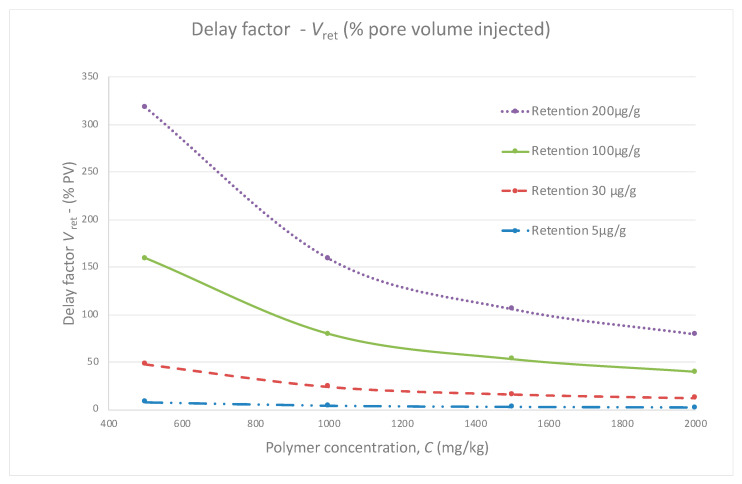
Examples of calculations of oil bank delay vs. retention and polymer concentrations values.

**Figure 2 polymers-13-02737-f002:**
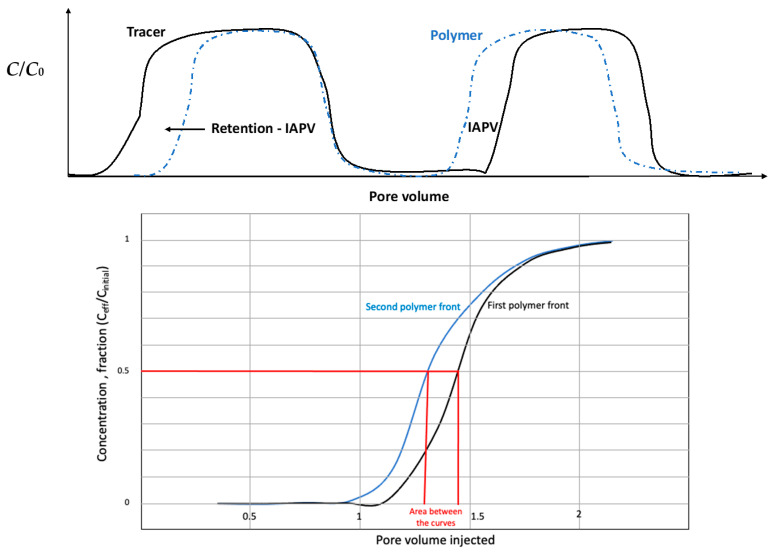
Example of retention calculation using the 2-fronts method. The first figure shows the overall principle of the method and how to calculate inaccessible pore volume. Once the polymer curves are normalized (*C*/*C*_0_ which is the ratio of measured concentration and initial concentration), the retention is calculated by considering the area between both curves (first & second fronts). The value is taken at the 0.5 cutoff for the ratio *C*_eff_/*C*_i_, being concentration in the effluents and initial concentration respectively.

**Figure 3 polymers-13-02737-f003:**
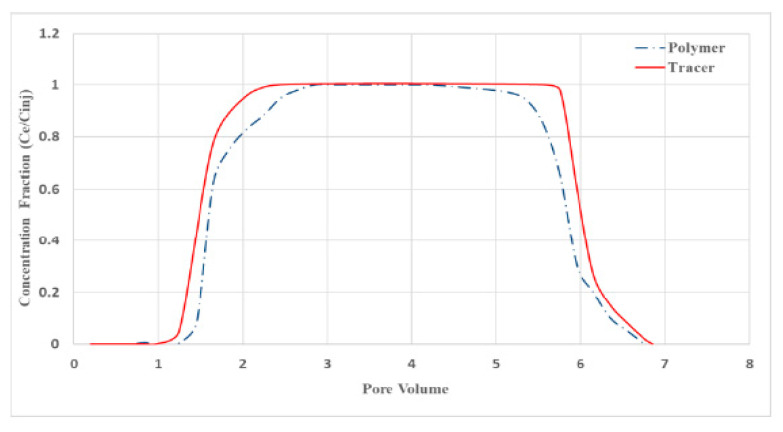
Single front method. Polymer and tracer are injected together and then flushed with water. The concentrations are analyzed during the process to determine the delay between tracer and polymer.

**Figure 4 polymers-13-02737-f004:**
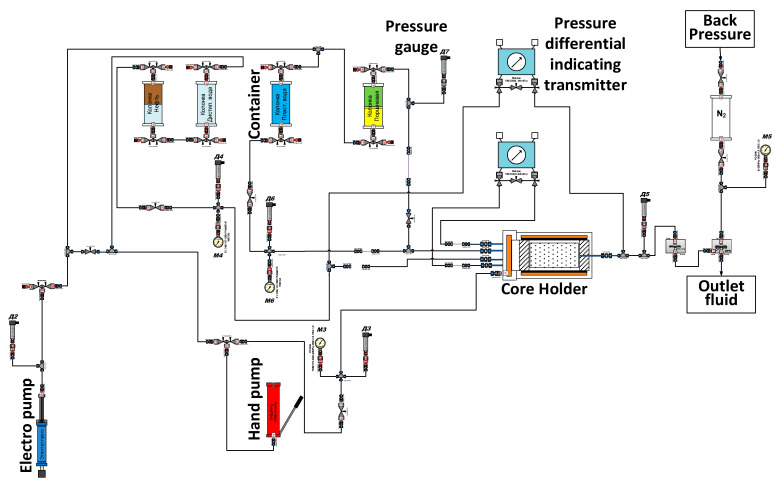
Hydraulic diagram of the Tech-ViP unit used for testing polymer compositions.

**Figure 5 polymers-13-02737-f005:**
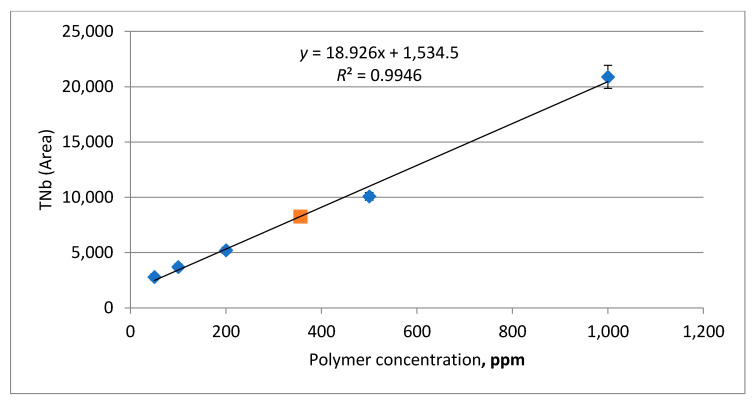
Electrochemical detector signal vs polymer A concentration, graduation dependence.

**Figure 6 polymers-13-02737-f006:**
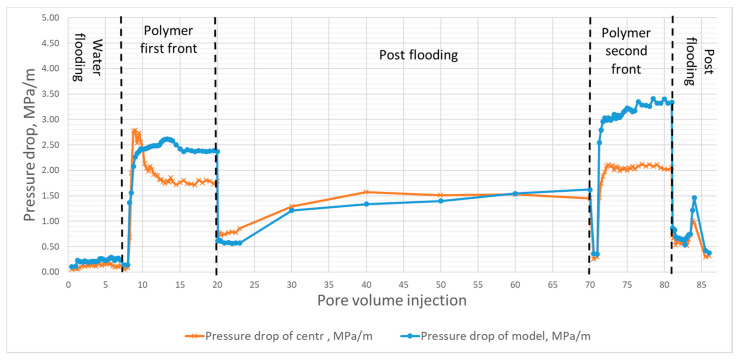
Pressure drops and polymer concentration changes during experiment No. 6 (core 1229 md, 0.01 Pa·s at 7.3 s^−1^ shear rate) as a function of pore volume injected (PV). A bump is noticed for pressure during the first polymer front (orange curve with the cross symbols or Grad center) which corresponds to a little volume of oil displaced by polymer, considered for the rest of the study.

**Figure 7 polymers-13-02737-f007:**
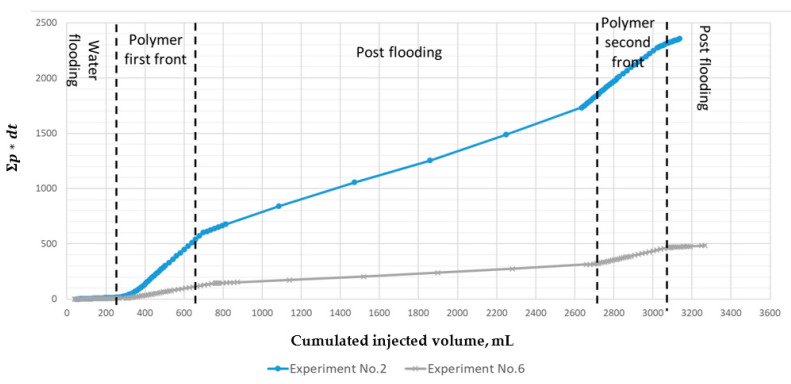
Hall plot for the entire core model at equal viscosities of the polymer solution (10 cP at 7.3 s^−1^) during experiments No. 2 and 6, upper and lower curves respectively. P stands for pressure while d*t* stands for time increment.

**Figure 8 polymers-13-02737-f008:**
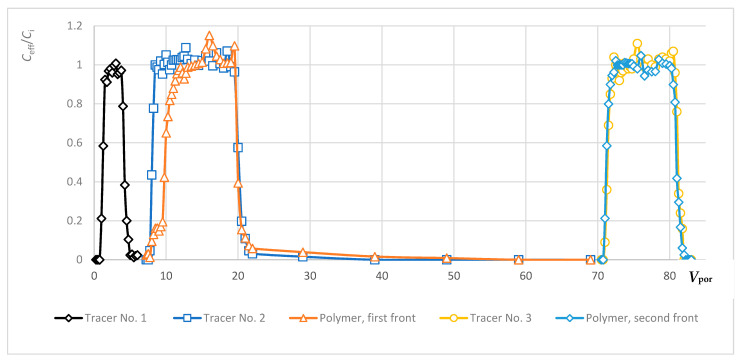
Normalized concentration (*C*_eff_/*C*_i_) of tracers and polymer in successive samples of the 6th filtration experiment. *C*_eff_ represents the measured concentration in the effluents at each sampling interval (each dot on the curves) and *C*_i_ represents the concentration of the fresh solution before injection.

**Table 1 polymers-13-02737-t001:** Brine composition for this study.

Salt	Concentration, g/L
NaHCO_3_	1.239
NaCl	14.118
KCl	0.592
MgCl_2_	0.594
CaCl_2_	0.833
Water salinity, g/L	17.38

**Table 2 polymers-13-02737-t002:** Information about core samples and polymers used. The polymers used for this study are exclusively polyacrylamides.

Experiment No.	Average Permeability, md (Helium)	Polymer	Mol. Weight Mean, MDa	Sulfate Degree, %	Concentration at 10 cP @ 7.3 s^−1^, ppm
1	346	A	18	-	900
2	361	A	18	-	900
3	424	B	12	-	1320
4	460	D	7	-	1250
5	552	C	12	5	1050
6	1229	C	12	5	1050

**Table 3 polymers-13-02737-t003:** Geological and physical characteristics of PK1-3 reservoir of the East-Messoyakhskoe field.

Formation	PK1-3
Temperature, °C	16
Rock pressure, MPa	19
Formation pressure, MPa	7.9
Water salinity, g/L	multicomponent composition
Oil density in atmospheric conditions, g/cm^3^	0.945
Oil density in formation conditions, g/cm^3^	0.922
Oil viscosity in formation conditions, MPa·s	111.15

**Table 4 polymers-13-02737-t004:** Slug volumes and injection rates for determination of retention using the dynamic 2-fronts method. The sequence is reproduced in the table, following the 2-fronts method described in Figure 2. The first line details the permeability of the samples. The second line summarizes the parameter for the first polymer + tracer front. The third line summarizes the parameter for the water flush between both fronts (experiments no. 2–6 has 2 rates in water injection, the first rate 2 m/day during 10 pore volumes, and the second rate 4 m/day during 40 pore volume). The fourth line details the injection parameters for the second polymer + tracer front. The last line details the parameters for the final water clean-up (experiments no. 2–6 has 2 rates in water injection, the first rate 2 m/day during 2 pore volumes, and the second rate 4 m/day during 1 pore volume).

Pumping Stages/Parameters		Experiment No.					
		1	2	3	4	5	6
Absolute permeability, md, average		346	361	424	460	552	1229
Polymer, first front	Linear injection rate, m/day	1	2	2	2	2	2
	Injected volume, PV	3	12	12	12	12	12
Formation water injection	Linear injection rate, m/day	2	2 (10PV) and 4 (40PV)	2 (10PV) and 4 (40PV)	2 (10PV) and 4 (40PV)	2 (10PV) and 4 (40PV)	2 (10PV) and 4 (40PV)
	Injected volume, PV	50	50	50	50	50	50
Polymer, second front	Linear injection rate, m/day	1	2	2	2	2	2
	Injected volume, PV	10	10	10	10	10	10
Formation water injection	Injected volume, PV	3	3	3	3	3	3
	Linear injection rate, m/day	1	2 (2PV) and 4 (1PV)	2 (2PV) and 4 (1PV)	2 (2PV) and 4 (1PV)	2 (2PV) and 4 (1PV)	2 (2PV) and 4 (1PV)

**Table 5 polymers-13-02737-t005:** Summary results of core studies.

Experiment No.	Polymer	Permeability, md	Mol. Mass, MDa	Sulfate Degree, %	Retention (Method 1), µg/g	Retention (Method 2), µg/g	Average Retention, µg/g
1	A	346	18	-	NA	NA	NA
2	A	361	18	-	444	436	440
3	B	424	12	-	385	321	353
4	D	460	7	-	152	292	222
5	C	552	12	5	124	290	207
6	C	1229	12	5	212	93	153

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
