# Peer review of "Polymer Retention Determination in Porous Media for Polymer Flooding in Unconsolidated Reservoir"

_polymers, 2021, doi:10.3390/polym13162737_

Round 1
Reviewer 1 Report
As such the current work has novelty. It is written in an engineering/report style and a bit at the edge of the typical style of Polymers. There are no major flaws. Following suggestions:
P 3 top three mechanisms; are reference available?
Polymers uses “Equation”. Please check.
Confused about: “Before testing, the polymer concentration was adjusted to a target viscosity of 10 cP 237 at 25 C.” Please explain.
Check y-axis Figure 5.
Overall the quality of the figures can be improved.
I would merge Results and Discussion sections; the current Discussion section is quite short.
Author Response
P 3 top three mechanisms; are reference available? Added one
Polymers uses “Equation”. Please check. Fixed
Confused about: “Before testing, the polymer concentration was adjusted to a target viscosity of 10 cP 237 at 25 C.” Please explain. Fixed
Check y-axis Figure 5. Fixed
Overall the quality of the figures can be improved.
I would merge Results and Discussion sections; the current Discussion section is quite short. Fixed
Reviewer 2 Report
The article falls only partially into my expertise scope, therefore I cannot evaluate responsibly its originality and significance.
However, it is evident that the article brings findings worth of sharing with other scientists, and it is only matter of the journal level and profile where they can be published.
In my opinion, Polymers is a journal where they can be published.
On the other hand, the text suffers from some imperfections to be fixed before acceptation.
In my opinion, the introduction is more factual and understandable than there is an average among manuscripts being submitted to Polymers, however not always proper references are stated here.
The similar applies to the other parts: The text itself is well arranged and understandable, but references are not always proper and formal flaws occur and tables and figures are insufficiently described and explained.
References format is inconsistent. Although using DOI is not obligatory and they are stated with many references, there are some items not easily identificable that have known DOI not involved in the list.
With more authors, the last one is sometimes divided by comma only, sometimes with "and", sometimes with "&". In my opinion, dividing all authors by comma is most reader-friendly.
Authors are divided mostly by commas, but sometimes by semicolons, between names and initials sometimes spaces only, sometimes commas.
In the text, references sometimes as superscript [4], mostly in the line level.
Italic and upright letters are not used properly for quantities and functions.
The IUPAP recommendation is available for example at https://iupap.org/wp-content/uploads/2014/05/A4.pdf.
The IUPAC recommended documents are https://iupac.org/wp-content/uploads/2019/05/IUPAC-GB3-2012-2ndPrinting-PDFsearchable.pdf
and https://iupac.org/wp-content/uploads/2016/01/ICTNS-On-the-use-of-italic-and-roman-fonts-for-symbols-in-scientific-text.pdf
Please, consider usage of recommended symbols.
There are a lot of abbreviations used in the text. Although at least some of them are properly explained at the first usage, involving their list would ease the reading, since it is quicker to check the list than to look for the first mention with an explanation.
lines 80-81: Ref. 7 is not about Canada, but it is connected with PJSXC Tatneft and Tatarstan; Ref. 8 is about Canada, not about Argentina.
line 109: In equation (1), variables should be in italics, not upright. Only the subscripts denoting material are correctly upright. Note that any correct equation has to be independent of units in which physical quantities are given, and possible coefficients with quantities may reflect in which units they are given. many readers will tend to interpret PV as product of pressure and volume (possible retention volume)
line 111: cm3 - 3 exponent should be in a superscript position: cm3
line 114: Figure 1 - Although the delay factor is a main subject of the figure, I do not see its definition in the text nearby (only of the retention value). It is not clear for a reader not delaying with a topic, whether polymer concentration is meant in the rock, in oil, or elsewhere. The curves are shrunk in the center of the figure; its rearrangement reducing empty areas and increasing the curves can be considered.
line 146: Figure 2 - C should be in italics as a physical quantity. I do not see an explanation of IAPV abbreviation (probably inaccessible pore volume).
line 212: According to the context, D means Darcy unit. Since it is not a SI unit and the article is intended for wider scientific public, mention at the first occurrence can be considered. My personal preference would be usage of SI units. In addition, not capital D, but lowercase d is used for this unit more often.
line 219: I am lost here what PK 1-3 reservoir is; probably, more readers would be, too.
line 225: Kp, Kpr used without an explanation
line 236: Molecular weights are stated, but it is not mention, whether there were monodisperse samples, or, otherwise, what type of molecular weight average is indicated
line 248: I appreciate including the hydraulic diagram. However, some texts are too small, additional comments and explanations would not be excessive.
line 266,269, 286: K is a physical quantity, therefore it should be in italics
line 297: In table 4, in some rows, (2PV) and similar means that the measured value is at double of retention volume, or what? The meaning of the table content should be exactly and clearly explained.
line 320: In Figure 5, physical quantities and variables are upright, while they should be in italics
line 343: What does "samples 0.1-0.5PV" mean? If "samples where 0.1 < PV < 0.5 ", its should be expressed correctly; if samples with values of something from 0.1 PV to 0.5 PV, it should be expressed again clearly.
line 346: In Figure 6, symbols of physical quantities (not the subscripts not denoting a variable) should be in italics. I do not see Vpor explanation, the legend does not specify what is on the horizontal axis.
line 376: In Figure 7, I do not see explained neither P nor T in the expression on the vertical axis. T denotes usually the temperature, but it seems to be time in this graph? Please, describe. Again, variablephysocal quantities and variables should be in italics, differentiation operator (if d is this case) upright.
line 393: In Figure 8, physical quantities and variables are upright, while they should be in italics. Subscript should be explained, the phrase "normalised dependence of concentration" is not sufficient.
line 529-532: References 2-4 incomplete, ref. 2 is probably https://doi.org/10.2118/196752-MS, ref. 3-4 parts of https://doi.org/10.2118/201822-MS
line 544, 547: In my opinion, full authors list in refs 9, 10 would be appropriate. Ref 10 has probably DOI https://doi.org/10.2523/IPTC-20285-MS
Author Response
Polymer retention determination in porous media for polymer flooding in unconsolidated reservoir
Reviewer 2 - fixed
|
Remarks |
Correction |
Comment |
|
With more authors, the last one is sometimes divided by comma only, sometimes with "and", sometimes with "&". In my opinion, dividing all authors by comma is most reader-friendly. Authors are divided mostly by commas, but sometimes by semicolons, between names and initials sometimes spaces only, sometimes commas. |
Fixed |
|
|
In the text, references sometimes as superscript [4], mostly in the line level. |
Fixed |
|
|
lines 80-81: Ref. 7 is not about Canada, but it is connected with PJSXC Tatneft and Tatarstan; Ref. 8 is about Canada, not about Argentina. |
Fixed |
|
|
line 109: In equation (1), variables should be in italics, not upright. Only the subscripts denoting material are correctly upright. Note that any correct equation has to be independent of units in which physical quantities are given, and possible coefficients with quantities may reflect in which units they are given. many readers will tend to interpret PV as product of pressure and volume (possible retention volume) |
Fixed |
|
|
line 111: cm3 - 3 exponent should be in a superscript position: cm3 |
Fixed |
|
|
line 114: Figure 1 - Although the delay factor is a main subject of the figure, I do not see its definition in the text nearby (only of the retention value). It is not clear for a reader not delaying with a topic, whether polymer concentration is meant in the rock, in oil, or elsewhere. The curves are shrunk in the center of the figure; its rearrangement reducing empty areas and increasing the curves can be considered. |
Fixed |
|
|
line 146: Figure 2 - C should be in italics as a physical quantity. I do not see an explanation of IAPV abbreviation (probably inaccessible pore volume). |
fixed |
|
|
line 212: According to the context, D means Darcy unit. Since it is not a SI unit and the article is intended for wider scientific public, mention at the first occurrence can be considered. My personal preference would be usage of SI units. In addition, not capital D, but lowercase d is used for this unit more often. |
Fixed |
|
|
line 219: I am lost here what PK 1-3 reservoir is; probably, more readers would be, too. |
fixed |
|
|
line 225: Kp, Kpr used without an explanation |
Fixed |
|
|
line 236: Molecular weights are stated, but it is not mention, whether there were monodisperse samples, or, otherwise, what type of molecular weight average is indicated |
Fixed |
|
|
line 248: I appreciate including the hydraulic diagram. However, some texts are too small, additional comments and explanations would not be excessive. |
Fixed |
|
|
line 266,269, 286: K is a physical quantity, therefore it should be in italics |
Fixed |
|
|
line 297: In table 4, in some rows, (2PV) and similar means that the measured value is at double of retention volume, or what? The meaning of the table content should be exactly and clearly explained. |
Fixed |
|
|
line 320: In Figure 5, physical quantities and variables are upright, while they should be in italics |
Fixed |
|
|
line 343: What does "samples 0.1-0.5PV" mean? If "samples where 0.1 < PV < 0.5 ", its should be expressed correctly; if samples with values of something from 0.1 PV to 0.5 PV, it should be expressed again clearly. |
Fixed |
|
|
line 346: In Figure 6, symbols of physical quantities (not the subscripts not denoting a variable) should be in italics. I do not see Vpor explanation, the legend does not specify what is on the horizontal axis. |
Fixed |
|
|
line 376: In Figure 7, I do not see explained neither P nor T in the expression on the vertical axis. T denotes usually the temperature, but it seems to be time in this graph? Please, describe. Again, variablephysocal quantities and variables should be in italics, differentiation operator (if d is this case) upright. |
Fixed |
|
|
line 393: In Figure 8, physical quantities and variables are upright, while they should be in italics. Subscript should be explained, the phrase "normalised dependence of concentration" is not sufficient. |
Fixed |
|
|
line 529-532: References 2-4 incomplete, ref. 2 is probably https://doi.org/10.2118/196752-MS, ref. 3-4 parts of https://doi.org/10.2118/201822-MS |
Fixed |
|
|
line 544, 547: In my opinion, full authors list in refs 9, 10 would be appropriate. Ref 10 has probably DOI https://doi.org/10.2523/IPTC-20285-MS |
Fixed |
|

Reviewer 3 Report
The work "Polymer retention determination in porous media for polymer flooding in unconsolidated reservoir" has an interesting and actual subject of the research.
The presented results are nice and useful, however:
- The structure of the work is not the most suitable,
- The first two sections should be restructured,
- The figures quality must be improved,
- The results section need a little amplified,
- The references section lost some valuable research, and must write at journal standards.
Author Response
Polymer retention determination in porous media for polymer flooding in unconsolidated reservoir
Reviewer 3 - fixed

Round 2
Reviewer 2 Report
I hoped that my comments to the revised version would be short. However, although many problems of original text have been fixed, the amount of persisting issues despite the declarations about their fixing leads me to commenting each item:
Original comment: References format is inconsistent. Although using DOI is not obligatory and they are stated with many references, there are some items not easily indentificable that have hown DOI not involved in the list.
With more authors, the last one is sometimes divided by comma only, sometimes with "and"m sometimes with "&". In my opinion, dividing all authors by comma is most reader-friendly.
Authors are divided mostly by commas, but sometimes by semicolons, between names and initials sometimes spaces only, sometimes commas.
Authors response: Fixed
Revised version comment: Although I appreciate adding DOI and other data to many items, inconsistency is not fixed despite the declaration. Supposing that the format used for refs 1-3 is what authors have chosen for this paper, ref. 4 has the extra "and" before the last author, refs. 4, 7, 8 and many other have commas between author surnames and initials; ref. 7, 10, 11, 12 etc. have the extra "&" before the last author` refs. 15 and 16 have commas between author surnames and initials, while authors are divided by semicolons. This is only example, not a full list of inconsistencies. It is a task for authors to fix the problem really.
Original comment: In the text, references sometimes as superscript [4], sometimes in the line level. Authors response: Fixed. Revised version comment: Seems to be all right.
Original comment: Italic and upright letters are not used properly for quantities and functions. Please, consider usage of recommended symbols.
Authors response: missing.
Revised version comment: Mostly corrected with a few exceptions. However, proprietary abbreviations are still used in the text instead of symbols recommended by IUPAC and IUPAP.
Original comment: There are a lot of abbreviations used in the text. Although at least some of them are properly explained at the first usage, involving their list would ease the reading, since it is quicker to check the list than to look for the first mention with an explanation.
Authors response: missing,
Revised version comment: List neither not added nor explicitly refused, finding explanation sometimes difficult
Original comment: lines 80-81: Ref. 7 is not about Canada, but it is connected with PJSXC Tatneft and Tatarstan; Ref. 8 is about Canada, not about Argentina.
Authors response: Fixed
Revised version comment: The references number in the text were corrected to correspond to correct items in the original version, however the numbers in the list have been shifted in the opposite direction, so that references in the text point again at wrong items.
Original comment: line 109: In equation (1), variables should be in italics, not upright. Only the subscripts denoting material are correctly upright. Note that any correct equation has to be independent of units in which physical quantities are given, and possible coefficients with quantities may reflect in which units they are given. many readers will tend to interpret PV as product of pressure and volume (possible retention volume)
Authors response: Fixed
Revised version comment: line 102: ret, rock, pret, polymer should be shifted to subscript position as it is correctly done in the text below – in comparison with the original version, the text after the equation has been corrected, but in the equation, correction of some symbols was accompanied by spoiling subscripts.
Original comment: line 111: cm3 - 3 exponent should be in a superscript position
Authors response: Fixed
Revised version comment: line 105 fixed.
Original comment: line 114: Figure 1 - Although the delay factor is a main subject of the figure, I do not see its definition in the text nearby (only of the retention value). ...
Authors response: Fixed
Revised version comment: Figure 1 is markedly better now. However, V should be in italics (Vret), as it is a physical quantity
Original comment: line 146: Figure 2 - C should be in italics as a physical quantity. I do not see an explanation of IAPV abbreviation (probably inaccessible pore volume).
Authors response: Fixed
Revised version comment: IAPV explained (line 134), however C has remained upright in the figure itself and line 151.
Original comment: line 212: According to the context, D means Darcy unit. Since it is not a SI unit and the article is intended for wider scientific public, mention at the first occurrence can be considered. My personal preference would be usage of SI units. In addition, not capital D, but lowercase d is used for this unit more often.
Authors response: Fixed
Revised version comment: All right (line 221).
Original comment: line 219: I am lost what PK 1-3 reservoir is; probably, more readers would be, too.
Authors response: Fixed
Revised version comment: Explanation should have been added rather in line 221 (of new version) – reformulation of the sentence to make clear that PK1-3 is the name of reservoir.
Original comment: line 225: Kp, Kpr used without an explanation
Authors response: Fixed
Revised version comment: Symbols have been removed from new line 236, I do not see in this moment whether they occur anywhere without explanation.
Original comment: line 236: Molecular weight are stated, but it is not mention, whether there were monodisperse samples, or, otherwise, what type of MW average is indicated
Authors response: Fixed
Revised version comment: Not exactly understood. Is it number average, weight average, viscosity average, z-average, or what type? (line 246)
Original comment: line 248: I appreciate including the hydraulic diagram. However, some texts are too small, additional comments and explanations would not be excessive.
Authors response: Fixed
Revised version comment: Some texts are still too small, but the important items are well readable now. The figure is acceptable, although there is still a possibility of an improvement.
Original comment: line 266,269, 286: K is a physical quantity, therefore it should be in italics
Authors response: Fixed
Revised version comment: All right.
Original comment: line 297: In table 4, in some rows, (2PV) and similar means that the measured value is at double of retention volume, or what? The meaning of the table content should be exactly and clearly explained.
Authors response: Fixed
Revised version comment: The meaning has been explained in the legend (lines 313-321). I feel the table still to be uncomfortable for reading and worth of appearance modification, but I do not insist on further change. Authors themselves may be interested in having their article reader friendly.
Original comment: line 320: In Figure 5, physical quantities and variables are upright, while they should be in italics
Authors response: Fixed
Revised version comment: In Figure 5, line 344, polymer concentration has been expanded into words and the description of vertical axis has been added, however symbols for physical quantities and variables have remained wrongly upright, while they should be in italics (y, R). TNb and TOC are examples of abbreviations used in the instrument manuals etc. instead of IUPAC and IUPAP recommended symbols of physical quantities. However, these abbreviations have been used in recent scientific articles so many times that it would not be fair to insist on abstaining from them just in this article.
Original comment: line 343: What does "samples 0.1-0.5PV" mean? If "samples where 0.1 < PV < 0.5 ", its should be expressed correctly; if samples with values of something from 0.1 PV to 0.5 PV, it should be expressed again clearly.
Authors response: Fixed
Revised version comment: Line 368 acceptable, although it could be yet clearer.
Original comment: line 346: In Figure 6, symbols of physical quantities (not the subscripts not denoting a variable) should be in italics. I do not see Vpor explanation, the legend does not specify what is on the horizontal axis.
Authors response: Fixed
Revised version comment: Expressed in full words in the revised version, it is better.
Original comment: line 376: In Figure 7, I do not see explained neither P nor T in the expression on the vertical axis. T denotes usually the temperature, but it seems to be time in this graph? Please, describe.
Authors response: Fixed
Revised version comment: Partially fixed, but P should be in italics (P), recommended symbol is p (lines 380, 406)
Original comment: line 393: In Figure 8, physical quantities and variables are upright, while they should be in italics. Subscript should be explained, the phrase "normalised dependence of concentration" is not sufficient.
Authors response: Fixed
Revised version comment: Vertical axis and legend corrected, V has remained upright at horizontal axis, although it should be in italics as the physical quantity (Vpor).
Original comment: line 529-532: References 2-4 incomplete, ref. 2 is probably https://doi.org/10.2118/196752-MS, ref. 3-4 parts of https://doi.org/10.2118/201822-MS
Authors response: Fixed
Revised version comment: All right
Original comment: line 544, 547: In my opinion, full authors list in refs 9, 10 would be appropriate. Ref 10 has probably DOI https://doi.org/10.2523/IPTC-20285-MS
Authors response: Fixed
Revised version comment: Corrected, but it is not clear which initial belongs to which name in Ref. 8
************
In addition, the abstract (second half) should be cleaned from misprints, in particular its version in the document metadata.
Line 20: Specify units in which ATBS content is expressed. Where in the text itself is this range specified?
Some of flaws mentioned can lead to misunderstanding some of the presented facts, therefore they should be fixed before an acceptation.
Author Response
Thank you for your time and review. Attached is a word file summarising the answers to your remarks in green.

Reviewer 3 Report
The work "Polymer retention determination in porous media for polymer flooding in unconsolidated reservoir" in the revised form should be considered for publish.
Some minor corrections on the references write are necessary.
Author Response
The corrections on the references have been made. Thank you for your time.